# Microbiological Properties of Microwave-Activated Carbons Impregnated with Enoxil and Nanoparticles of Ag and Se

**Oleg Petuhov [1],\*, Tudor Lupascu [2], Dominika Behunová [3], Igor Povar [2], Tatiana Mitina [2] and Maria Rusu [2]**

[1]   Ecosorbent Limited Liability Company, 51, Dacia str., MD-2062 Chisinau, Moldova
[2]   Institute of Chemistry, 3, Academiei str., MD-2028 Chisinau, Moldova; lupascut@gmail.com (T.L.); ipovar@yahoo.ca (I.P.); mitina_tatiana@mail.ru (T.M.); maria_rusu_md@yahoo.com (M.R.)
[3]   Institute of Geotechnics SAS, Watsonova, 45, 040 01 Košice, Slovakia; behunova@saske.sk
\*   Correspondence: petuhov.chem@gmail.com; Tel.: +373-22-72-54-90

**Abstract:** Microwave-activated carbons from walnut shells (ACMW) were impregnated with Ag and Se nanoparticles and with the Enoxil biologically active preparation, and the microbiological properties of the obtained composites were studied. To increase the functionality of the adsorbent, the activated carbon was oxidized with ozone, resulting in ACMWO containing aliphatic and aromatic carboxylic groups. There was a considerable decrease in the specific surface of the activated carbon after the oxidation process. Nitrogen adsorption was used to determine the structural parameters of the activated carbons. A simultaneous thermal analysis was used to study the thermal behavior of intact and oxidized activated carbons. Infrared spectroscopy was applied to analyze the surface chemistry of the adsorbents. The microbiological activity of the activated carbons was studied using *Escherichia coli* bacteria and *Candida albicans* fungi. The kinetic study of the microbiological activity allowed the estimation of the bactericidal/fungicidal action time of the activated carbons.

**Keywords:** activated carbon; microwaves; nanoparticles; adsorption; microbiological properties

## 1. Introduction

Activated carbons (ACs) are widely used in various fields, such as for water purification [1–3], in the pharmaceutical industry, for human body detoxification and external wound processing [4,5], and as catalytic and antimicrobials supports. Recently, activated carbons have started to be used in clinical medicine, in absorbent dressings for superficial and profound lesions, as well as for post-operative trauma restauration. These bandages contribute to the creation of a favorable environment for rapid wound healing and do not allow the spread of infections. Activated carbons intensify the hemocoagulation process, enhance exudate accumulation on the surface, create a barrier for microorganism penetration from the outside and, in combination with the dressing material, prevent wound dehydration. A more pronounced biological activity is obtained by impregnating the surface of activated carbons with biologically active substances [6]. The bactericidal activity of silver and selenium nanoparticles is well known and documented in many papers [7–9], and the Enoxil preparation has pronounced antimicrobial properties as well [10]. Enoxil is a mixture of monomeric derivatives of catechin and epicatechin, in their free form and esterified with gallic acid, and peroxidic compounds. This preparation shows high antioxidant activity and enhanced therapeutic properties [11].

Currently, ACs are obtained from cheap and renewable raw materials by methods requiring low energy consumption [12]. A perspective research line in this context is the use of microwaves in the

process of ACs preparation. The employment of microwaves is finding a widening application in various fields because of the many advantages of microwave heating in comparison with traditional methods, such as fast heating, lack of inertia in heat transfer, and activation time reduction, which in turn lead to diminished consumption of energy and activation agents [13].

The oxidation of ACs aims at the formation of various functional groups on their surface for a stronger interaction with nanoparticles. Acid and base functional groups are also formed during the production of ACs, depending on the activation methods employed. However, the amount of these groups is very small, mainly because of their instability at the elevated temperatures at which the activation process takes place. The oxidation process can occur at the interaction of AC with liquid chemical agents ($HNO_3$, $H_2O_2$, $(NH4)_2S_2O_8$, $K_2Cr_2O_7$) or gases ($O_2$, $O_3$, $Cl_2$) [14–16]. The advantages provided by gaseous ozone oxidation cannot be achieved by using other oxidizing agents. The process takes place at room temperature, the structure of the activated carbon is not contaminated with heteroatoms as a result of oxidation, functional groups of a certain type are formed, and the surface of the AC is sterilized during the process [17]. These features make it possible to obtain activated carbons suitable for use in medicine.

The purpose of this study was to obtain mesoporous activated carbon with low ash content, impregnate this adsorbent with the Enoxil medicinal preparation and with Ag and Se nanoparticles, qualitatively and quantitatively analyze the immobilization processes of neutral nanoparticles on intact and oxidized activated carbons, and test the microbiological activity of the obtained samples.

## 2. Materials and Methods

Walnut shells of local origin were used as the raw material for obtaining AC. Phosphoric acid (85 wt %) of analytical grade was used as the activating agent.

### 2.1. Preparation of Activated Carbon

Microwave-activated carbons from walnut shells (ACMW) was obtained by microwave treatment of walnut shells impregnated with phosphoric acid. Initially, walnut shells of 3.15–5.0 mm in size were dried for 4 h at 110 °C and then they were impregnated with 85% phosphoric acid, the ratio of walnut shells weight to phosphoric acid volume being of 1:2. Immediately afterwards, the nut shells and phosphoric acid mixture was transferred in a quartz reactor and subjected to activation in a microwave oven for 5 min at a power of 700 W. The obtained activated carbon was fractionated, washed with distilled water until the pH of the wash solution reached the value 6.0, and dried at 110 °C to constant mass.

### 2.2. Activated Carbon Ozonizing

Modification of activated carbon chemistry was performed by passing an ozone-containing gas stream through a glass reactor containing the 90–125 μm fraction of the ACMW carbon. Ozone was obtained in an oxygen-fed ozone generator. The flow rate was 300 $cm^3$/min, the ozone mass fraction was 10%. The ozonizing process was performed for 30 min at 20 °C with continuous stirring of the carbon powder with a magnetic stirrer. Due to the fact that the process is exothermic, the vessel with activated carbon was thermostated. The AC was dried at 110 °C for 4 h after oxidation and stored for further analysis. As a result, the ACMWO was obtained, which was studied by various physicochemical methods and also used in further microbiological studies after impregnation with Ag, Se nanoparticles, and Enoxil. To study the reversibility of ozone chemosorption and find out to what extent the activated carbon structure had changed, the ACMWO sample was heated in vacuum at 400 °C for 24 h, obtaining the ACMWO-400 sample.

### 2.3. Thermal Analysis

Thermal analysis was performed on the Derivatograph Q-1500D system (MOM, Mateszalka, Hungary). The samples (about 20 mg) were placed in a platinum crucible and heated from

20 to 1000 °C under a nitrogen flow rate of 100 mL/min and a heating rate of 10 °C/min. Thermogravimetric (TG), derivative weight loss (DTG) and differential thermal analysis (DTA) curves were registered simultaneously.

## 2.4. Gas Adsorption Characterization

The structure and adsorption parameters of activated carbons were obtained from nitrogen adsorption isotherms at 77 K. The adsorption isotherms were measured using Autosorb-1MP (Quantachrome Instruments, Boynton Beach, FL, US) with prior degassing at 110 °C for 12 h. Samples impregnated with Ag and Se nanoparticles and the Enoxil-impregnated sample were degassed at 40 °C before analysis. The specific surface area ($S_{BET}$) was calculated using the Brunauer–Emmett–Teller (BET) equation. The total pore volume ($V_t$) was calculated by converting the amount of nitrogen gas adsorbed at a relative pressure of 0.99 to the equivalent liquid volume of the adsorbate. The volume of micropores ($V_{mi}$) was determined using the t-method, the volume of mesopores ($V_{me}$) was determined from the difference between the total volume and the volume of micropores. The Dubinin–Radushkevich (DR) method was used to calculate the adsorption energy ($E_a$). Non-local density functional theory (NLDFT) based on a slit-shaped pore equilibrium model was used for the calculation of the pore volume distribution as a function of radius and effective radius ($r_{ef}$). The isotherms were collected and analyzed using AS1Win software version 2.01.

## 2.5. Infrared Spectroscopy (IR)

Fourier-Transform Infrared Spectroscopy (FTIR) was used to determine the functional groups of activated carbons. IR spectra of activated carbons were recorded by the FT-IR Spectrum 100 installation (PerkinElmer Inc., Wellesley, MA, US) in the range of 4000–650 cm$^{-1}$. FT-IR spectra were collected using Perkin Elmer software Spectrum version 6.2.0.

## 2.6. Impregnation of Activated Carbons with Biologically Active Substances

The impregnation of ACMW and ACMWO was performed by mixing the samples, pre-dried and weighed, with a 5% Enoxil aqueous solution or a Poviargol preparation containing 1% silver nanoparticles and selenium nanoparticles stabilized with bovine serum, the nanoparticle mass fraction being 0.1%. Poviargol is a finely dispersed silver metal stabilized by low-molecular-weight medical polyvinylpyrrolidone. Metallic silver in Poviargol is in the form of spherical nanoclusters with a narrow particle size distribution in the range of 10–40 nm, with the bulk of particles (more than 80%) being silver particles of 10–20 nm in size. The sizes of elemental selenium particles range from 30 to 80 nm, the average size being 50 nm. The mixture was kept for 24 h at room temperature, then it was filtered, washed with distilled water, and dried at room temperature over $P_2O_5$. The samples thus obtained, i.e., ACMW-E, ACMWO-E (impregnated with Enoxil), ACMW-Ag, ACMWO-Ag (impregnated with silver nanoparticles), and ACMW-Se, ACMWO-Se (impregnated with selenium nanoparticles) were stored for quantitative analysis of the impregnated components and for subsequent microbiological tests.

## 2.7. Microbiological Tests

The antibacterial activity of the samples was tested using the method of counting colony-forming units (CFUs) that resisted to the action of the studied preparation. The test sample with a 20 mg mass was passed into double-distilled water (10 mL) containing approximately $10^3$ CFU/mL of *Escherichia coli* (ATCC 25922) or $10^2$ CFU/mL of *Candida albicans* fungi (ATCC 10231). The mixture was aerobically incubated at 20 °C with continuous stirring. At predetermined time intervals (10, 30, 60, 240, 420, and 1440 min), 0.1 mL of the solution was taken and passed into Petri dishes with agar medium to assess the colonization capacity of the organic nutrient substrate. The Petri dishes were thermostated at 37 °C for 24 h, after which the formed colony units were photographed and counted. The exact determination of the initial concentration of the microorganisms and the follow-up of the external factors was done by measuring the control samples containing no active carbon under the same conditions, taking

0.1 mL of the solution at the same time and assessing the colonization capacity. All measurements were duplicated, the fixed value being the arithmetic mean of the parallel analyses.

## 3. Results and Discussion

### 3.1. Structural Parameters of Activated Carbons

The structural and adsorption parameters of intact and ozone-treated ACs are presented in Table 1. ACMW is an adsorbent with a mixed pore structure, predominantly mesoporous. The adsorption isotherms of nitrogen on the activated carbon samples are shown in Figure 1. All isotherms contain hysteresis rings indicating a mesoporous structure of activated carbons. The form of the isotherms is similar for all samples, indicating preservation of the morphology for the oxidized sample, and, at the same time, the adsorption volume decreases considerably. AC ozonization resulted in a sudden decrease of the specific surface from 1369 to 172 $m^2/g$. The same trend of decreasing values was also observed for pore volume.

**Table 1.** Structural and adsorption parameters of intact and oxidized activated carbons.

| AC | $S_{BET}$, $m^2/g$ | $S_{me}$, $m^2/g$ | $V_{mi}$, $cm^3/g$ | $V_{me}$, $cm^3/g$ | $V_t$, $cm^3/g$ | $E_a$, kJ/mol | $r_{ef}$, Å |
|---|---|---|---|---|---|---|---|
| **ACMW** | 1369 | 788 | 0.260 | 1.360 | 1.566 | 12.9 | 10.2 |
| **ACMWO** | 172 | 111 | 0.029 | 0.259 | 0.288 | 6.4 | 12.4 |
| **ACMWO-400** | 530 | 229 | 0.157 | 0.459 | 0.616 | 14.1 | 6.4 |
| **ACMW-Ag** | 1062 | 627 | 0.224 | 1.009 | 1.233 | 11.3 | 6.4 |
| **ACMW-E** | 1188 | 763 | 0.216 | 1.208 | 1.424 | 11.3 | 6.4 |

ACMW: Microwave-activated carbons from walnut shells; ACMWO: ozonized ACMW, ACMWO-400: ACMWO heated at 400 °C under vacuum, ACMW-Ag: Ag-treated ACMW, ACMW-E: Enoxil-treated ACMW, $S_{BET}$: specific surface area (determined by Brunauer–Emmett–Teller (BET) equation), t: total pores, me: mesopores, mi: micropores, $E_a$: adsorption energy, $r_{ef}$: effective radius.

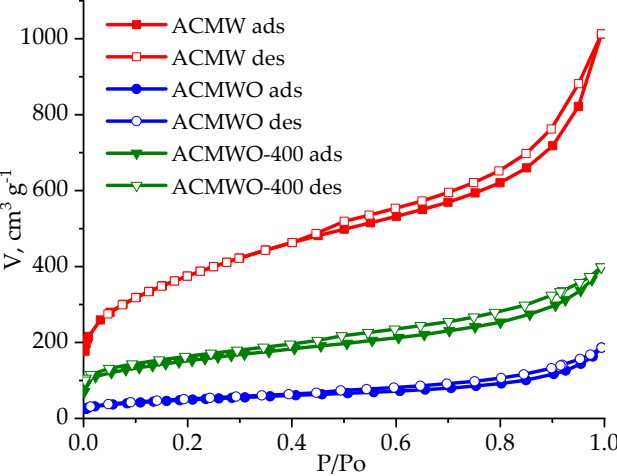

**Figure 1.** Adsorption–desorption isotherms of nitrogen on intact ACMW (microwave-activated carbons from walnut shells) and ACMW oxidized with ozone.

It is important to note that the pore volume change was proportional, irrespective of the type of pores, i.e., the oxidation did not occur selectively, and the decrease of the total volume of the meso- and micropores was 82%, 81% and 89%, respectively, resulting in a homogeneous distribution of the functional groups throughout the surface. The same conclusion could be made by analyzing the pore volume distribution curves shown in Figure 2. It is observed that the curve profiles are similar. At the same time, for oxidized AC, there was a relative decrease in pore volume, proportional over the entire interval. A literature analysis indicated a similar behavior of carbonic substances in the ozonization process, i.e., a significant decrease in the specific surface and pore volume [18–21].

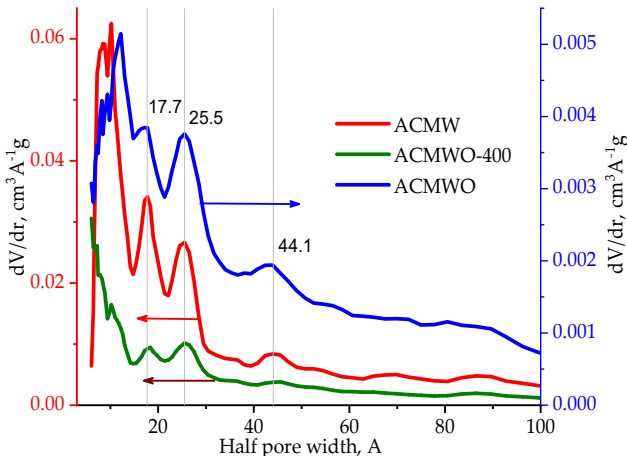

**Figure 2.** Pore radii distribution of intact ACMW and ACMW oxidized with ozone.

At the same time, the explanation of these effects is not univocal, and various assumptions can be made: the formation and elimination of volatile compounds in the ozonization process [19], the destruction of the pore walls [18,20], the blocking of the pores by formation of functional groups [18].

By comparing the pore distribution curves of the three analyzed samples, shown in Figure 2, we observed a similarity in the mesoporous domain and significant changes in the microporous domains, r < 10 Å. In all distribution curves, maximum values for pores with radii of 18, 25, and 44 Å are observed, their ratio remains the same, and only their absolute values change.

This led to the conclusion that the volume of mesopores decreased due to the formation of functional groups, blocking the access of nitrogen molecules, and their morphology remained unchanged. Micropores blocking was also observed, as well as their enlargement from 10 to 12 Å during the oxidation process, by partial AC oxidation and the formation of volatile compounds. Vacuum heating up to 400 °C of the oxidized sample led to the elimination of more thermally unstable groups, so the peak in the 10–12 Å region completely disappeared. Impregnation with Ag nanoparticles and the Enoxil preparation of the ACMW resulted in a decrease in the specific surface of ACs by 22% and 13%, respectively, as shown in Table 1. Figure 3 shows the adsorption–desorption isotherms of nitrogen on ACMW-Ag and ACMW-E that indicate a small decrease in the volume of adsorbed nitrogen compared to the initial sample (Figure 1).

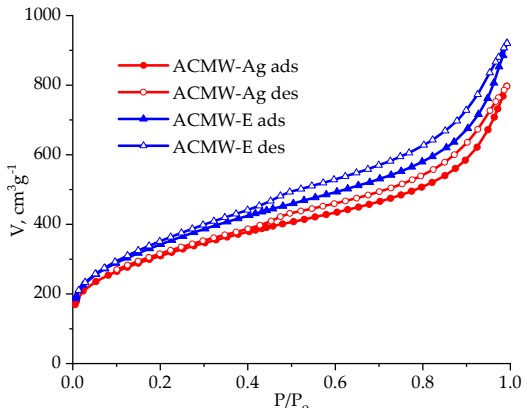

**Figure 3.** Adsorption–desorption isotherms of nitrogen on ACMW-Ag and ACMW-E.

The pore volume distribution curves indicated an essential decrease of mesopores with the radius of 17 Å and a shift of the peak for the pores with the radii of 44 to 38 Å, as shown in Figure 4 (compare with Figure 2). This is explained by the deposition of Ag nanoparticles and the Enoxil preparation predominantly in mesopores, reducing the pore size.

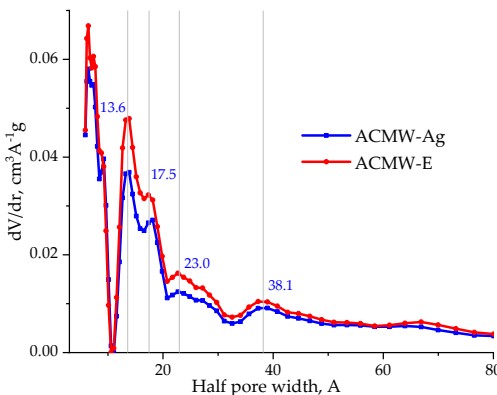

**Figure 4.** Pore radii distribution of ACMW-Ag and ACMW-E.

## 3.2. Thermal Analysis

In order to estimate the thermal stability of the oxidation-based functional groups and to observe the reversibility of the chemosorption process, a thermal analysis of the intact and oxidized samples was performed, and the results are shown in Figure 5.

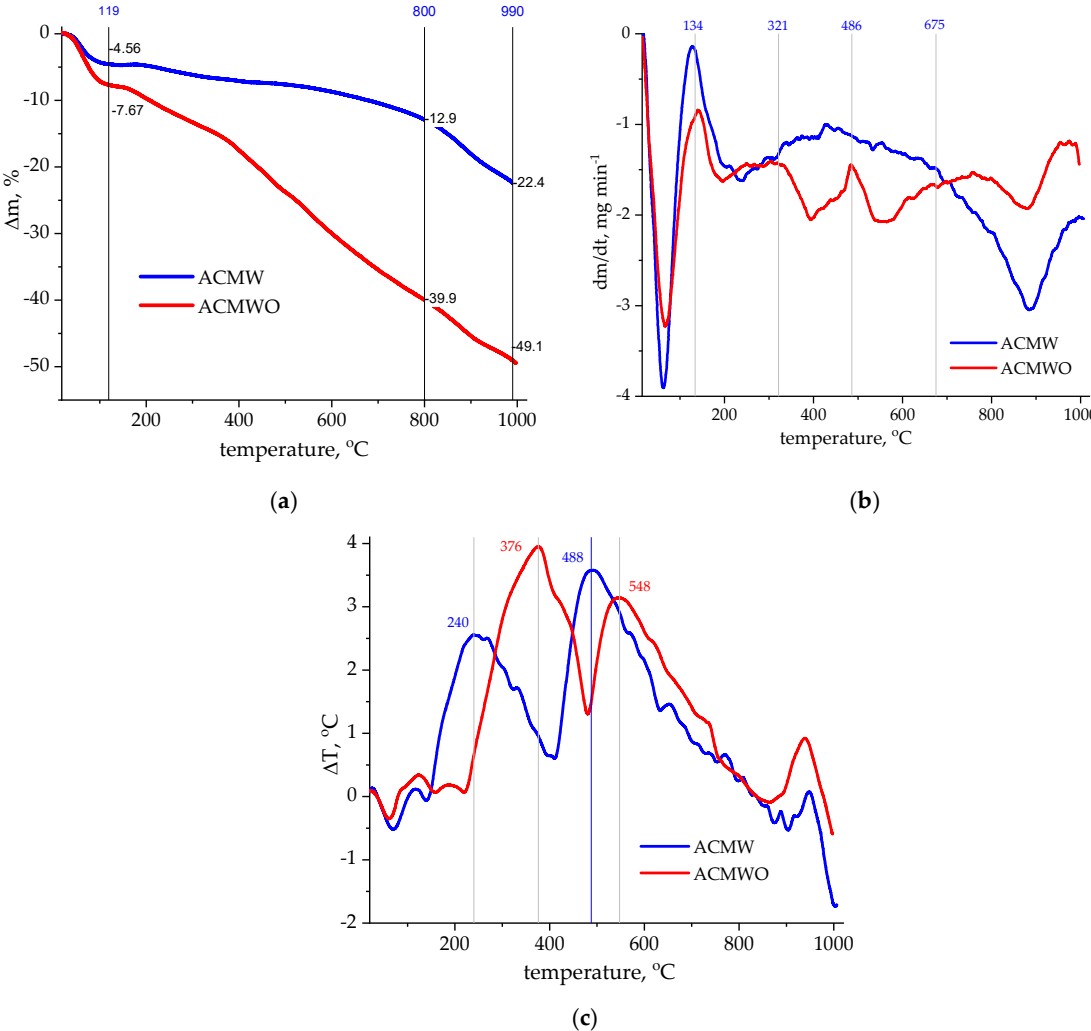

**Figure 5.** Thermoanalytical curves of the intact and oxidized ACMW: (**a**) thermogravimetric (TG), (**b**) derivative weight loss (DTG) and (**c**) differential thermal analysis (DTA).

In the temperature range of 20–120 °C, the adsorbed water was eliminated, the loss of mass for the oxidized AC being 60% higher due to the pronounced hydrophilic character of the AC as a result of oxidation. As it can be seen from the DTG curves, the profiles of both curves are similar up to 320 °C, indicating the same thermal processes, namely, the elimination of water molecules from the meso- and micropores. Starting at 320 °C, a distinctive decomposition was observed on the DTG curve of the oxidized activated carbon, with the maximum decomposition observed at 395 °C. In the range of 486–675 °C, a new characteristic transformation was observed, with a maximum at 548 °C. The shape of both curves over this range indicates complex processes, resulting from the overlapping of several elementary stages [22]. After 800 °C, a similar thermal degradation took place up to 1000 °C, with a loss of mass of 10% in both cases, caused by the transformation of the phosphoric acid residues and the carbonaceous shell. The process was accompanied by a weak endothermic effect with a maximum at 865 °C. Since phosphorus compounds in the activated carbon structure have a similar thermal behavior, one can say that they were not affected by the ozonization process, suggesting that the phosphorus groups were passive and were not dangerous in contact with a biological environment.

Data from the literature indicate the possibility of formation of various functional groups on the surface of activated carbon in the ozonation process: carboxylic, carbonyl, phenolic, lactonic groups [23–25]. Their elimination in the process of thermal treatment results in the formation of $CO_2$ from the carboxylic and lactonic groups or CO following the decomposition of the phenolic, quinonic, and etheric fractions [20,26]. Because of the different thermal stability of the functional groups, an approximate estimate of the character of these groups could be made, knowing that CO was the predominant product eliminated within the range of 500–800 °C. Figure 5c shows the existence of at least two types of functional groups, the first groups were thermally stable up to 490 °C, while the second groups remained present even up to 700 °C.

This explained the partial restoration of the specific surface of the oxidized activated carbon when heated under vacuum at 400 °C.

The correlation of the results obtained from the thermal analysis and nitrogen adsorption indicated that the ozonization process led to the formation of functional groups of predominantly acidic nature on the surface of activated carbon; the oxidation took place homogeneously throughout the surface, leading to the blocking of pores and a considerable reduction of the specific surface; ozone chemosorption was partially reversible after desorption at elevated temperatures leading to incomplete restoration of the specific surface, while the morphological structure of the adsorbent underwent insignificant changes.

*3.3. Infrared Spectroscopy Analysis*

Infrared analysis of the studied activated carbons samples allowed a more detailed description of the AC surface chemistry. The infrared spectra in Figure 6 indicate major changes resulting from the ozonization process.

Figure 6 shows the IR spectra of the intact and oxidized ACMW. The broad band in the domain of 870–750 $cm^{-1}$ was attributed to off-plane vibrations of C–H bonds in aromatic systems, containing substituents at different positions of the benzene ring [27,28] or furanic acid [29]. The asymmetric stretching vibration of the C–H bond from the methylene group was confirmed by the presence of a peak at 2900 $cm^{-1}$. The maximum at 1699 $cm^{-1}$ was attributed to C=C bonds stretching vibrations with an aliphatic character [27]. The intense peak at 1559 $cm^{-1}$ was attributed to the C=C bond stretching vibrations in the benzene rings [30].

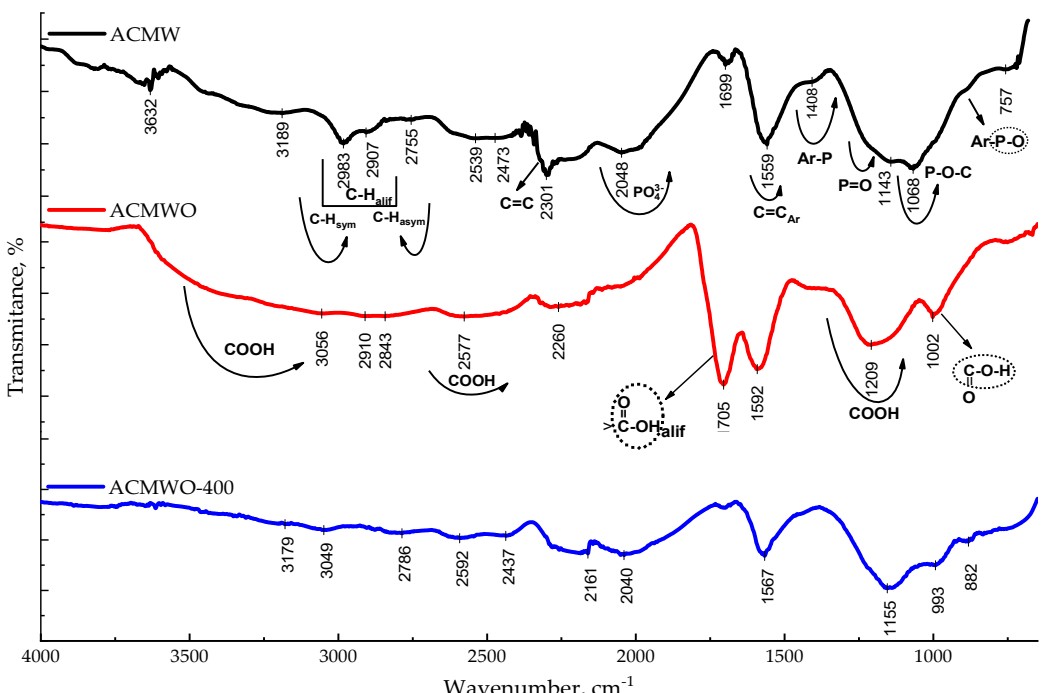

**Figure 6.** Infrared spectra of intact and modified activated carbons.

Phosphoric acid and its esters were identified by the broad band at a maximum of 2040 cm$^{-1}$ [25]. Aliphatic phosphorus compounds containing P–O–C bonds were identified by the presence of a very wide and intense band with a maximum of 1068 cm$^{-1}$, characteristic of asymmetric stretching vibrations [27,31,32]. The shoulder at 1228 cm$^{-1}$ was attributed to the P=O link stretching vibration from the aliphatic groups [33].

Absorption bands characteristic of phosphorus-containing groups linked to the benzene ring have a lower intensity than the bands assigned to aliphatic compounds. At the same time, the formation of phosphorus bonds with the benzene ring was suggested by the presence of band maxima at 890 cm$^{-1}$ (P–O) and 1406 cm$^{-1}$ (Ar–P) [27,34].

The adsorption bands present in the oxidized AC spectrum were attributed to the vibrations of the C=O (1705 cm$^{-1}$) and C–O (1002 cm$^{-1}$) bonds of the aliphatic carbon-linked groups, while the band peak at 1209 cm$^{-1}$ was attributed to the vibrations of the C=O bonds of the benzene ring-linked carboxyl groups [24,27,30,34–37]. The broad adsorption bands with a maximum of 2577 cm$^{-1}$, as well as the existence of a wide band without a maximum defined in the region of 2500–3300 cm$^{-1}$, also indicated the presence of carboxylic groups in the structure [30].

The treatment of the oxidized activated carbon at 400 °C led to the reduction or even the disappearance of adsorption bands, and the IR spectrum became very similar to that of the original AC, as shown in Figure 4. Firstly, the peaks of the aliphatic carboxyl groups disappeared, and, simultaneously, the band in the region of 1203 cm$^{-1}$ became less intense. This led to the conclusion that the thermal treatment led to the selective decarboxylation of the carboxylic groups, except for the aromatic ones which were thermally more stable. These results complement the observations made on the basis of thermal analysis, which also showed the presence of functional groups with different thermal stability.

The combination of gas adsorption methods, thermal analysis, and IR spectroscopy allowed the elucidation of the mechanism of modification of the activated carbon chemistry that occurred at ozonation. It was found that ozone is a strong oxidizing agent that interacts with activated carbon at room temperature and leads to the formation of carboxyl groups linked to the aliphatic and aromatic carbon atoms. The ozone chemosorption process considerably diminished the specific surface area

of activated carbons by forming functional groups in the pores and blocking the access of nitrogen molecules. At the same time, this process was partially reversible—the thermal treatment allowed for the destruction of the formed groups and a partial restoration of the pore volume.

To determine the content of nanoparticles retained on the surface, the activated carbons were calcined for 6 h at 500 °C by the addition of a mixture of sulfuric and nitric acids to prevent the volatilization of the analyzed components. Table 2 presents the results of atomic adsorption spectroscopy analysis of solutions obtained by dissolving the calcination residue in a sulfuric and nitric acid mixture.

**Table 2.** Content of silver and selenium in the initial and modified activated carbons. ACMW-Se: Se-treated ACMW, ACMWO-Se: se-treated ACMWO.

| Sample | C(Ag), mg/g | C(Se), µg/g |
|---|---|---|
| ACMW | <0.001 | <0.05 |
| ACMWO | <0.001 | <0.05 |
| ACMW-Ag | 1.48 | <0.05 |
| ACMWO-Ag | 1.02 | <0.05 |
| ACMW-Se | <0.001 | 17.14 |
| ACMWO-Se | <0.001 | 2.97 |

As it can be seen, the ACMWO possessed an adsorption capacity of Ag nanoparticles close to that of the original sample, even with a specific surface area eight times smaller. This is explained by the influence of functional groups on the immobilization of the nanoparticles. A more obvious difference in the retention capacity was found in the adsorption of selenium nanoparticles, which was six times less for the oxidized sample, indicating an insignificant contribution of the functional groups.

### 3.4. Microbiological Tests of Activated Carbons

Microbiological analyses of the activated carbon samples impregnated with the Enoxil preparation, Ag, or Se nanoparticles were performed using the ATCC-25922 Gram-negative bacteria *E. coli* and the ATCC-10231 fungal strains *C. albicans*. The method of analysis we chose allowed the kinetic study of the action of the preparations on the cultures and the determination of the time when the preparations exhibited bactericidal properties. Photographs confirming the growth of microorganisms' colonies and kinetic curves of sample activity are shown in Figures 7 and 8, respectively.

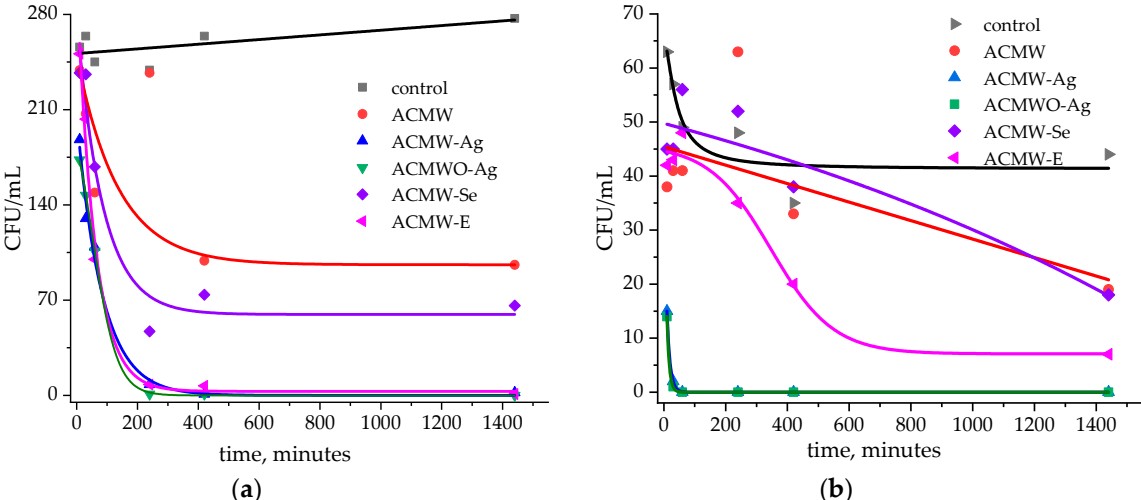

**Figure 7.** Kinetics of the microbiological activity of activated carbons on *Escherichia coli* (**a**) and *Candida albicans* (**b**).

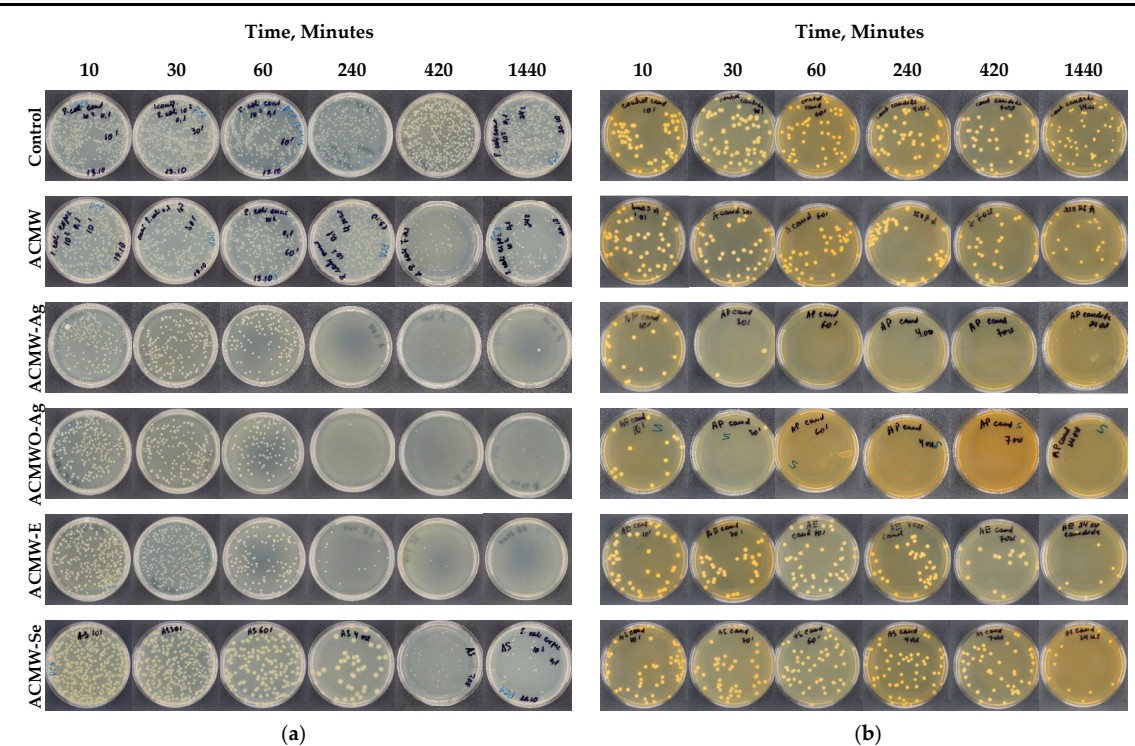

**Figure 8.** Residual *E. coli* (**a**) and *C. albicans* (**b**) after contact with the activated carbons.

As can be seen, the number of CFU of *E. coli* in the control sample increased over 24 h, indicating a lack of influence of external factors and increased resistance of the bacteria over time. At the same time, the concentration of *C. albicans* decreases in the absence of the nutrient medium by 30% in 24 h.

In the presence of intact activated carbon there was a 50% decrease in bacteria and fungi over 24 h, which is explained by their adsorption on the surface of the AC. Although there was a decrease in the concentration of the microorganisms, unimpregnated activated carbon cannot be used for medicinal purposes because it becomes itself a medium with high bacterial and fungal content.

Very good results were obtained for the active carbons samples impregnated with Ag nanoparticles, but also for those impregnated with the Enoxil preparation. Thus, as seen in Figure 7a, the concentration of *E. coli* decreased to zero after 4 h of contact. By comparing ACMW-Ag and ACMWO-Ag carbons, we observed a similar bacteriological activity towards *E. coli* and *C. albicans*. Such a result indicates a major influence of Ag nanoparticles on bactericidal properties, while the adsorption process remains of secondary importance.

These results allowed the establishment of the bacterial action mechanism of AC impregnated with Ag nanoparticles: the bactericidal activity of the nanoparticles took place in the solution by gradual elimination of the nanoparticles from the surface of the activated carbon and interaction with bacteria and fungi; the value of the specific surface area of the activated carbon had a secondary role, however, the surface chemistry influenced the amount of adsorbed nanoparticles and their rate of elimination into the surrounding environment, as shown in Figure 9.

The Enoxil-impregnated active carbon sample exhibited bactericidal properties after 4 h in *E. coli* and inhibited the development of *C. albicans*. The ACMW-Se had no apparent bactericidal or fungicidal activity, and its kinetic curves were similar to those of intact AC, so that the reduction in microorganism concentrations was due only to the adsorption process. The best results towards *C. albicans* were shown by ACMW-Ag and ACMWO-Ag, with the bactericidal effect confirmed after 60 min of contact. Even after 10 min, there was a major decrease in fungi cultures, as shown in Figure 8b.

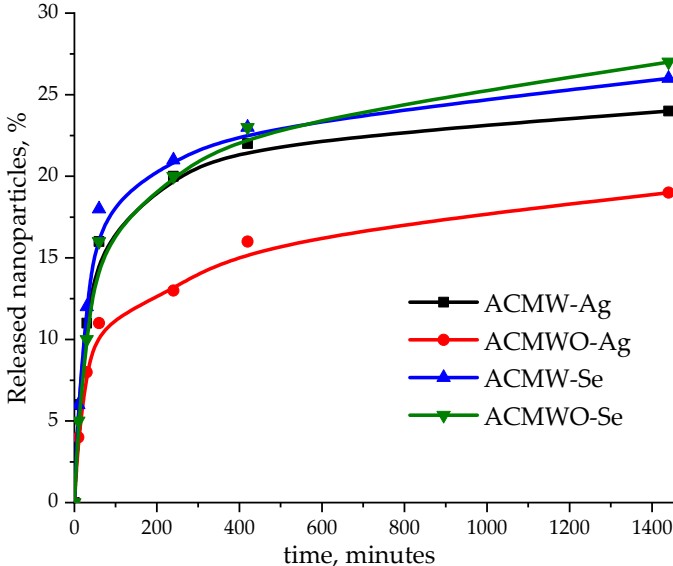

**Figure 9.** Kinetic curves for the removal of Ag and Se nanoparticles from the surface of activated carbons.

The obtained results suggest the following range of microbiological activity towards *E. coli* and *C. albicans*: ACMWO-Ag = ACMW-Ag > ACMW-E > ACMW-Se = ACMW.

To study the rate of deposition of the nanoparticles on the surface of impregnated activated carbon, the concentrations of silver and selenium in solution were measured at the same time intervals, as shown in Figure 9. As it can be seen, the rate of removal of Ag nanoparticles from the oxidized activated carbon sample was about 30% lower than that measured for Se.

## 4. Conclusions

In this study, activated carbon from nut shells was obtained by microwave activation and was further oxidized and impregnated with Ag and Se nanoparticles and the biologically active preparation Enoxil. Microwave activation allowed obtaining mesoporous ACMW carbon with a high specific surface area and a low mineral content over a short period of time. The oxidation of the obtained AC resulted in the sterilization of the adsorbent and the formation of acidic functional groups on their surface, while the oxidation process produced a decrease of the specific surface from 1369 to 172 $m^2$/g. Ozone oxidation was shown not to destroy the structure of the carbon, and the chemosorption process was partially reversible on heating. Impregnation of intact and oxidized nanoparticles allowed obtaining adsorbents that exhibited increased microbiological activity. A pronounced bactericidal effect was shown by samples impregnated with Ag nanoparticles and the Enoxil preparation, which showed their bactericidal activity against *E. coli* in 4 h, destroying the microorganism cultures. The fungicidal activity of Ag-impregnated carbons was observed after one hour of contact with *C. albicans*. The Enoxil preparation showed fungal inhibition activity after 24 h. The intact activated carbon led to a decrease in the concentration of microorganisms as well, but in the absence of a bactericidal/fungicidal preparation, this increases the risk of conserving and spreading the microorganisms. The obtained results suggest the use of activated carbons impregnated with nanoparticles and Enoxil preparation in various medical fields, in particular for the creation of bactericidal dressings that would protect the penetration of infections within the body, comfortably keeping of humidity in the area of open wounds, with adsorption and subsequent destruction of the infected exudate.

**Author Contributions:** Investigation, methodology, visualization and writing—original draft, O.P.; project administration, T.L.; investigation, D.B.; investigation, T.M.; investigation, M.R.; writing—review & editing, I.P.

**Funding:** Authors would like to acknowledge financial support from H2020 (MSCA-RISE-2016/NanoMed Project, grant number 734641).

**Conflicts of Interest:** The authors declare no conflict of interest. The funders had no role in the design of the study; in the collection, analyses, or interpretation of data; in the writing of the manuscript, or in the decision to publish the results.

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
