# Peer review of "Microbiological Properties of Microwave-Activated Carbons Impregnated with Enoxil and Nanoparticles of Ag and Se"

_carbon_

Round 1
Reviewer 1 Report
Please see the attachment

Author Response
Dear Reviewer,
thank you very much for the analysis and helpful comments on my manuscript.
Considering your comments, the required corrections and changes have been made. Also, the whole manuscript was edited for improving the grammar quality.
Best regards,
Petuhov Oleg.

Reviewer 2 Report
The authors show experimental results on activated carbon obtained by microwave activation, which was subsequently oxidized and impregnated with Ag and Se nanoparticles as well as biologically active Enoxil. The authors propose that microwave activation leads to mesoporous ACMW carbon with high specific surface area. By performing ozone oxidation, the sterilization of the adsorbent and the formation of acidic functional groups on their surface are produced, while the oxidation process causes an important decrease of the specific surface. One the most interesting contributions of this work to the bactericidal dressings field is based on the pronounced bactericidal effect of samples that have been impregnated with Ag nanoparticles and Enoxil, which showed their bactericidal activity against Escherichia coli .The fungicidal activity of Ag-impregnated carbons is also shown with Candida albicans.
I find that the present study is of great interest, however, I strongly suggest a mayor revision of the English. It is hard to read and understand most of the explanations due to the English quality.
Regarding the introduction, employed methods and obtained results, I have some general questions:
1. Introduction: references concerning the effect of using ozono in the sterilization process as a previous stage to understand the microbiological tests are missing.
2. Sample nomenclature: it is quite repetitive. For example: ACMW is used for microwave activated carbons. Thus, the use of phrases such as “activated carbon ACMW” is excessive.
3. Methods: Previous to the microbiological tests, explain or clarify whether any sterilization protocol was employed, in order to clearly correlate the final obtained results with the determination of the initial concentration of the microorganisms.
4. Results and discussion: the authors state in lines 158-160 that “the volume of mesopores decreases due to the formation of functional groups and blocking the access of the nitrogen molecules, their morphology remaining unchanged “. A clearer explanation about which are the expected functional groups and why the morphology is unchanged is needed, in order to better understand this partial conclusion.
5. In lines 167 and 168, the authors state: “This is explained by the deposition of Ag nanoparticles and the Enoxil preparation predominantly in mesopores”. A clearer discussion of Figure 4 is necessary to better understand this statement.
6. Figure 3 is not mentioned and explained in the main text.
7. Lines 185 and 186: “shape of both curves over this range indicates complex processes, resulting from the overlapping of several elementary stages”. References are missing.
8. Line 191: “do not pose a danger at contact with the biological environment.” References are missing
9. Ag and Se NPs: Is it possible to estimate the mean size of the NPs? An STM, AFM or SEM image would be helpful.
10. Line 261: “do not pose a danger at contact with the biological environment.” References are missing.
11.Conclusions: which is the thermal and temporal stability of the composites?
Author Response

(The authors gave the same response as above.)

Round 2
Reviewer 2 Report
The authors have improved the manuscript.